# Region-specific H3K9me3 gain in aged somatic tissues in *Caenorhabditis elegans*

Cheng-Lin Li[1], Mintie Pu[2], Wenke Wang[1], Amaresh Chaturbedi[1], Felicity J. Emerson[1], Siu Sylvia Lee[1]*

**1** Department of Molecular Biology and Genetics, Cornell University, Ithaca, New York, United States of America, **2** State Key Laboratory for Conservation and Utilization of Bio-Resources and Center for Life Science, School of Life Sciences, Yunnan University, Kunming, Yunnan, China

\* sylvia.lee@cornell.edu

**Data Availability Statement:** All ChIP-seq data are available in the NCBI BioProject repository (accession number PRJNA699886). (https://www.

## Abstract

Epigenetic alterations occur as organisms age, and lead to chromatin deterioration, loss of transcriptional silencing and genomic instability. Dysregulation of the epigenome has been associated with increased susceptibility to age-related disorders. In this study, we aimed to characterize the age-dependent changes of the epigenome and, in turn, to understand epigenetic processes that drive aging phenotypes. We focused on the aging-associated changes in the repressive histone marks H3K9me3 and H3K27me3 in *C. elegans*. We observed region-specific gain and loss of both histone marks, but the changes are more evident for H3K9me3. We further found alteration of heterochromatic boundaries in aged somatic tissues. Interestingly, we discovered that the most statistically significant changes reflected H3K9me3-marked regions that are formed during aging, and are absent in developing worms, which we termed "aging-specific repressive regions" (ASRRs). These ASRRs preferentially occur in genic regions that are marked by high levels of H3K9me2 and H3K36me2 in larval stages. Maintenance of high H3K9me2 levels in these regions have been shown to correlate with a longer lifespan. Next, we examined whether the changes in repressive histone marks lead to de-silencing of repetitive DNA elements, as reported for several other organisms. We observed increased expression of active repetitive DNA elements but not global re-activation of silent repeats in old worms, likely due to the distributed nature of repetitive elements in the *C. elegans* genome. Intriguingly, CELE45, a putative short interspersed nuclear element (SINE), was greatly overexpressed at old age and upon heat stress. SINEs have been suggested to regulate transcription in response to various cellular stresses in mammals. It is likely that CELE45 RNAs also play roles in stress response and aging in *C. elegans*. Taken together, our study revealed significant and specific age-dependent changes in repressive histone modifications and repetitive elements, providing important insights into aging biology.

ncbi.nlm.nih.gov/Traces/study/?acc=
PRJNA699886).

**Funding:** This work was supported by R01 grant
AG024425 from the National Institute on Aging
(https://www.nia.nih.gov/) and the Cornell
University Center for Vertebrate Genomics (https://
cvg.cornell.edu) seed grant to SSL. The funders
had no role in study design, data collection and
analysis, decision to publish, or preparation of the
manuscript.

**Competing interests:** The authors have declared
that no competing interests exist.

## Author summary

Heterochromatin refers to the portion of the genome that is tightly packed where genes
stay silent. Heterochromatin is typically decorated by particular chemical groups called
histone modifications, such as trimethylation of lysine 9 or lysine 27 on histone 3
(H3K9me3 or H3K27me3). To understand how the heterochromatin landscape may
change from a "youthful" to an "aged" state, we monitored the genome-wide patterns of
H3K9me3 and H3K27me3 during aging using the genetic model soil worm *C. elegans*. We
found that while H3K27me3 remained relatively stable with age, H3K9me3 showed sub-
stantial gain and loss at specific loci in aged worms. We observed that new H3K9me3-
marked heterochromatin preferentially formed in specific gene-rich regions in aged
worms. Interestingly, these particular regions were marked by high levels of three other
histone modifications when worms were young. This result suggested that H3K9me3 gain
during aging is influenced by the gene-specific landscape of histone modifications estab-
lished at young age rather than that it occurs in a stochastic manner. In summary, our
study discovered reproducible and gene-specific changes in histone modifications that
likely contribute to the aging phenotypes.

## Introduction

In eukaryotes, heterochromatin is the condensed portion of the genome that is typically
located at the nuclear periphery and is transcriptionally repressed [1]. The association between
aging and dysregulated repressive heterochromatin have been observed across species, from
yeast to humans [2–8]. In yeast, loss of transcriptional silencing occurs during replicative
senescence and contributes to age-related sterility [2–4]. In flies, heterochromatin levels posi-
tively correlate with fly lifespan; Breakdown of heterochromatin at old age contributes to de-
repression of ribosomal RNAs and genome instability [5]. In worms, aging is associated with
loss of peripheral heterochromatin and reduced levels of repressive histone modifications
[6,9,10]. In mice, the expression of repetitive DNA elements progressively increases in senes-
cent cells and aging somatic tissues [7,11]. In humans, loss of heterochromatin-associated
markers and increased expression of repetitive DNA elements are hallmarks of aging in cul-
tured cells from normally aged individuals and from patients with Hutchinson-Gilford proge-
ria syndrome (HGPS) [12–14]. Taken together, the deterioration of heterochromatin structure
and loss of transcriptional repression are characteristic features of aging.

 Heterochromatic regions are typically decorated with one of the two evolutionarily con-
served histone modifications, tri-methylation on histone H3 at lysine 9 (H3K9me) and tri-
methylation on histone H3 at lysine 27 (H3K27me3). In general, H3K9me3 is associated with
constitutive heterochromatin, which covers repeat-rich, gene-poor regions near centromeres
and telomeres [15]. H3K9me3 is implicated in suppressing endogenous retrovirus and repeti-
tive DNA elements [16,17]. H3K27me3, on the other hand, is localized at facultative hetero-
chromatin, which forms in gene-rich regions and conditionally represses target genes in a
cellular context-dependent and DNA sequence-specific manner. H3K27me3 is involved in
silencing developmental genes and X-chromosome inactivation [18,19]. Previous studies have
reported both aging-associated gain and loss of repressive histone modifications (H3K9me3
and H3K27me3). For instance, a global decrease in H3K9me3 and H3K27me3 levels were
observed during aging in whole worms and human cells [9,10,13,20,21]. In flies, the levels of
H3K9me3 were found to decrease in the aging intestine [22] but to increase in the aging brain
[23]. In mice, the levels of H3K27me3 reduced in senescent fibroblast cells [20] but elevated in

the brain of a mouse model of accelerated aging [24]. Therefore, it remains unclear how age-associated changes in H3K9me3 and H3K27me3 levels contribute to heterochromatin deterioration and the aging phenotypes. It has been proposed that it is not the net abundance but rather the genomic distribution and utilization of the histone modifications that regulate the aging process [25].

*Caenorhabditis elegans* lacks cytologically dense-staining heterochromatin due to its holocentric chromosomes with dispersed centromeres. Repressive H3K9me3 and H3K27me3 in worms are predominantly localized on repeat-enriched distal arms of all five autosomes, where their distributions are largely overlapped [26–28]. In contrast, heterochromatin is mostly marked by either H3K9me3 or H3K27me3 in monocentric species, such as flies and humans [28]. Moreover, H3K27me3-marked regions that lack H3K9me3 can be found in the central regions of autosomes and on the X chromosome in worms. In this study, we investigated how H3K9me3 and H3K27me3 change during aging in the somatic tissues of germlineless *glp-1* mutants by chromatin immunoprecipitation (ChIP) followed by deep sequencing. We found site-specific loss and gain of H3K9me3 and H3K27me3 in aged somatic cells. Age-dependent loss of repressive marks was predominantly found in the central regions of autosomes while gain was localized on the distal autosomal arms. Furthermore, region-specific gain of H3K9me3 at old age resulted in the formation of new H3K9me3-marked heterochromatic regions, which we termed "aging-specific repressive regions" (ASRRs). These ASRRs were preferentially localized at genic regions marked by H3K9me2 and H3K36me1/2 at the juvenile stages. We hypothesized that the formation of ASRRs was a result of H3K9me2-to-H3K9me3 conversion and that the trimethylation of H3K9me2 could contribute to structural and functional changes of chromatin at old age. We next examined whether somatic aging in the *glp-1* mutants is associated with loss of transcriptional suppression of repetitive DNA elements. Although we did find an increase in the levels of RNAs derived from repetitive DNA elements at old age, that nevertheless represented a small fraction (<1%) of the overall transcriptome. Interestingly, we found CELE45, a short interspersed nuclear element (SINE), to be significantly over-represented among the repeats with elevated expression in aged worms. In murine and human cells, SINE RNAs were induced and implicated in cellular stress response [29–31]. To this end, we re-analyzed the publicly available RNA-seq data and revealed that CELE45 was induced upon heat stress in worms. Moreover, our analyses indicated that over-expression of CELE45 in the aged germlineless *glp-1* mutants was not due to the culture temperature of 25°C and/or lack of germline. These results indicated that CELE45 was specifically induced in response to aging.

## Results

### Deterioration of H3K9me3 boundaries in aged somatic tissues in *C. elegans*

To examine the genome-wide patterns of repressive histone modifications (H3K9me3 and H3K27me3) in young and aged somatic tissues of *C. elegans*, we performed chromatin-immunoprecipitation followed by sequencing (ChIP-seq) in germlineless worms (S1 Table). We used *glp-1(e2141)* mutants that lack germ cells at the non-permissive temperature (25°C) to harvest post-mitotic somatic tissues and prepare whole worm extracts [32]. For both repressive histone marks, we had two ChIP-seq biological replicates in each of the two time points (day 2 and day 12 of adulthood). We chose day 2 as the young time point when wild-type worms normally reach peak reproduction, and day 12 as the old time point when 10–20% of the *glp-1 (e2141)* population would have died [33]. To compare the similarity of the ChIP-seq replicates, we performed pair-wise Pearson's correlation analysis on H3, H3K9me3, and H3K27me3 ChIP-seq tag coverage in 15kb sliding windows along the genome (S1 Fig). Biological

replicates for the two repressive marks and H3 control at the same age had genome-wide correlation coefficients higher than 0.83, indicating high reproducibility.

Next, we identified H3K9me3- and H3K27me3-enriched peak regions normalized to the control H3, using the broad peak parameters in MACS2 (v2.1.0). We combined all biological replicates for peak calling and merged peaks identified at both time points (day 2 and day 12 adults) in order to generate common regions for downstream age-dependent analyses. We noticed a number of gap regions between two broad peaks showed positive fold enrichment (log2) of repressive marks relative to H3 at both time points (S2A Fig). In these cases, we reasoned that the two neighboring peaks represent one long enrichment domain (S2A Fig). We merged them into a single broad peak and the threshold gap size was empirically determined (S2B and S2C Fig). This last step was intended to more accurately capture the broad nature of the H3K27me3 and H3K9me3 marking. The final peak sets had 12,516 H3K9me3 peak regions (S2 Table), and 5,870 H3K27me3 peak regions (S3 Table). The H3K9me3 peak regions were predominantly enriched in the repeat-rich distal arms of autosomes (S2 Table and S3A Fig). H3K27me3 coincided with H3K9me3 on the distal autosomal arms but was also enriched on X chromosome and in the H3K9me3-depleted central regions of autosomes (S3 Table and S3A Fig). The genome-wide distribution of both repressive marks in germlineless glp-1 mutants are similar to those from wild-type L3 larvae as previously published (S3A Fig) [26,27,34]. Indeed, correlation analysis showed significant positive correlations for both H3K9me3 (Pearson's correlation coefficient = 0.52, p-value < 2.2e-16) and H3K27me3 peak profiles (Pearson's correlation coefficient = 0.38, p-value < 2.2e-16) between our ChIP-seq data in glp-1 adults and published data in wild-type L3 larvae [28,34]. To further evaluate the peak calling results, we compared the distribution of repressive peak regions with our published ChIP-seq data of active histone marks (H3K4me3 and H3K36me3) [33,35]. As expected, the genome-wide distributions of active and repressive marks were largely mutually exclusive (S3B Fig). The mutually exclusive occupancy was most evident between H3K27me3 and H3K36me3 (Pearson's correlation coefficient = −0.71), similar to earlier observations [36,37].

The H3K9me3 and H3K27me3 repressive histone marks are generally associated with transcriptional repression. To confirm this, we used our previously published RNA-seq data of day 2 and day 12 glp-1(e2141) germlineless adults [33] to assess the expression of the genes whose coding sequences overlap with either H3K9me3 or H3K27me3 peak regions. As expected, the majority of these genes showed no detectable RNA expression at both the young and old time points (64.2% and 73.6%, respectively). Moreover, genes associated with the repressive peak regions showed significantly lower RNA levels compared to genes in non-peak regions (S4A and S4B Fig). In contrast, the majority of the genes that overlap with our published peak regions of active marks (H3K4me3 and H3K36me3) in glp-1 adults were actively expressed in at least one time point (69.2% and 78.2%, respectively) [33,35].

To examine whether there were age-dependent changes in H3K9me3 and H3K27me3 enrichment at ChIP-seq peak regions in germlineless glp-1 animals, we first visualized the similarity between young and old samples by performing multidimensional scaling (MDS) analysis. The MDS plots showed that H3K9me3 and H3K27me3 ChIP-seq experiments were separated and clustered by age (Fig 1A and 1B). In comparison, H3 ChIP-seq profiles were highly similar between the two ages (Fig 1A and 1B) in germlineless glp-1 animals as previously reported [33,35].

Next, we examined how the average H3K9me3 and H3K27me3 signals at ChIP-seq peak regions changed between young and old glp-1 mutants. We included our published H3K4me3 and H3K36me3 ChIP-seq data in the analysis for comparisons [33,35]. At H3K9me3 peak regions, we found an overall reduced H3K9me3 enrichment at old age, and concomitant changes for the active histone marks H3K4me3 and H3K36me3 at the boundaries. In young

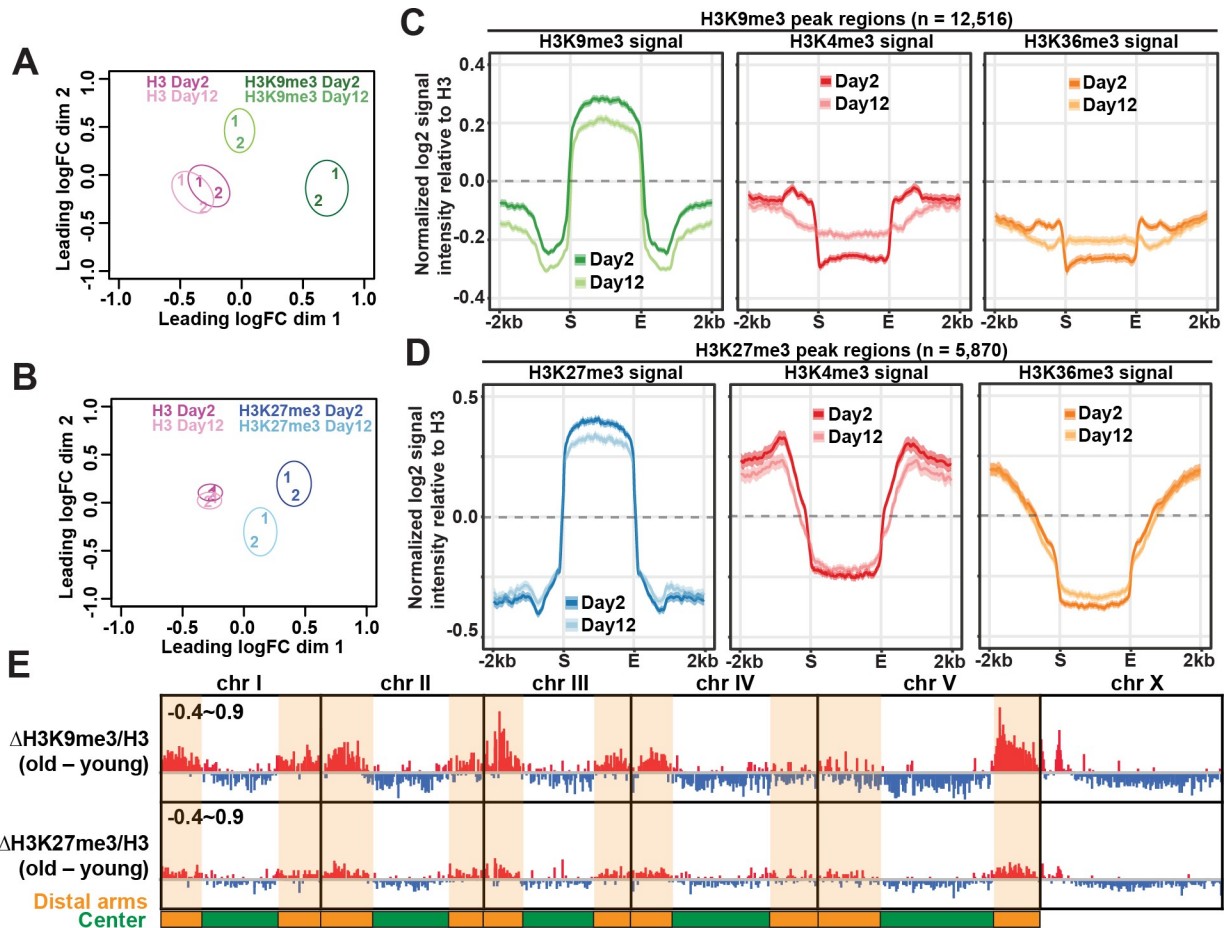

**Fig 1. Region-specific gain and loss of repressive marks in aged somatic tissues.** Multidimensional scaling (MDS) plots showing the similarity between ChIP-seq replicates of H3 controls (A & B), H3K9me3 (A), and H3K27me3 (B) in young (day 2 adults) and old (day 12 adults) worms. Two biological replicates are represented by numbers (1 and 2). There were age-dependent changes in H3K9me3 and H3K27me3 but not H3 controls. (C) Metaplots showing the average z-scores of H3K9me3, H3K4me3 and H3K36me3 ChIP-seq signal intensity normalized to H3 ($log_2$) across H3K9me3 peak regions (± 2kb flanking regions; n = 12,516). The average H3K9me3 enrichment across all peaks was reduced with age. The sharp decline of H3K4me3 and H3K36me3 at the boundaries of H3K9me3 peaks was concomitantly lessened with age. (D) Metaplots showing the average z-scores of H3K27me3, H3K4me3 and H3K36me3 ChIP-seq signal intensity normalized to H3 ($log_2$) across H3K27me3 peak regions (± 2kb flanking regions; n = 5,870). The average H3K27me3 enrichment across all peaks was reduced with age. (C-D) The solid lines represent the average z-scores with shaded areas showing the 95% confidence intervals. (E) IGV browser view showing age-dependent changes (old–young) in the average z-scores of normalized H3K9me3 and H3K27me3 signal intensity along the entire genome in 20-bp bins. Red and blue bars indicate an increase and decrease, respectively, in the average z-scores. The numbers on the top left corner of each track indicate the y-axis range. Distal arms of autosomes are highlighted by the orange shaded regions, and central regions are shaded green.

somatic tissues, H3K9me3 peak regions coincided with a sharp fall in the levels of these active histone marks (H3K4me3 and H3K36me3), exhibiting clear boundaries (Fig 1C). This distinctive feature became blurred in aged somatic tissues, indicating invasion of active marks in H3K9me3-marked constitutive heterochromatin during aging. At H3K27me3 peak regions, we also observed decreased H3K27me3 enrichment, but the more gradual decline of active histone marks at the boundaries remained stable with age (Fig 1D). It was known that the mutually exclusive patterns of H3K27me3 and H3K36me3 were maintained by the antagonistic relationship between Polycomb repressive complex 2 (MES-2/3/6) and H3K36 methyltransferase (MES-4). We found that the RNA expression of the four genes remained stable during aging [33], which would be consistent with our observation that the H3K27me3 and H3K36me3 boundaries were preserved in aged *glp-1* germlineless mutants.

## Region-specific gain and loss of H3K9me3 and H3K27me3 in aged somatic tissues in *C. elegans*

Like other eukaryotes, the *C. elegans* genome is organized into transcriptionally active and inactive chromatin compartments in the nucleus [38,39]. We next examined whether spatially partitioned heterochromatic chromosome arms and euchromatic central regions show distinct age-dependent changes in H3K9me3 and H3K27me3 enrichment in *glp-1* animals. Interestingly, age-dependent loss in H3K9me3 and H3K27me3 mainly occurred in the active central regions of autosomes and most of the X chromosome (Figs 1E and S5A–S5C). In contrast, age-dependent H3K9me3 and H3K27me3 gain was mostly observed in the heterochromatic regions, including the distal arms of all five autosomes and the left arm of the X chromosome (Figs 1E, S5B and S5D). Whereas the H3K9me3 gain in heterochromatic arms was highly visible, and the average signal intensity across all peaks in distal arms indicated statistical significance between young and old animals, that of H3K27me3 was relatively modest and insignificant (Figs 1E and S5D). In sum, euchromatic central regions became further depleted of both repressive histone marks while heterochromatic distal arms become significantly more enriched for H3K9me3 in aged somatic tissues in *glp-1* mutants.

To identify the peak regions with significant changes in H3K9me3 and H3K27me3 with age in the *glp-1* mutants, we performed differential analysis using edgeR [40]. We found that H3K9me3 peak regions had more statistically significant changes with age (578 upregulated and 17 downregulated peaks; 4.7% of all peaks; S4 Table and S6A Fig) than H3K27me3 peak regions (59 upregulated and 7 downregulated peaks; 1.1% of all peaks; S5 Table and S6B Fig). We noticed a disproportionately large fraction of the H3K9me3 and H3K27me3 differential peak regions gained repressive marks with age (578 of 595 H3K9me3 differential peaks and 59 of 66 H3K27me3 differential peaks; S6A and S6B Fig). Most of the peak regions that lost repressive H3K9me3 or H3K27me3 marks were not statistically significant at FDR cutoff of 0.05. While H3K9me3 peak regions substantially overlapped with H3K27me3 peak regions (~80% of the area), age-dependent changes in H3K9me3 and H3K27me3 largely occurred in distinct domains. On average, only 6.6% of upregulated and 12.7% of downregulated peak regions of both repressive marks overlapped.

Similar to S4 Fig, we next divided the repressive peak regions into three groups based on their age-dependent changes (upregulated, downregulated, no significant change) and associated them with gene expression levels based on our previous RNA-seq data [33]. We found that genes overlapping with the non-significant and upregulated repressive peak regions showed significantly lower RNA expression levels compared to non-peak regions (Fig 2A and 2D) as discussed earlier in S4 Fig. Genes in the upregulated H3K9me3 peak regions had significantly higher RNA levels than genes in non-significant H3K9me3 peak regions at both young and old time points (KS test *p-value* < 0.0001; Fig 2A and 2B), suggesting the genomic regions that showed age-dependent gain in H3K9me3 are likely transcriptionally active. In contrast, RNA levels were lower in genes associated with the upregulated H3K27me3 peak regions than non-significant H3K27me3 peak regions (KS test *p-value* = 0.013; Fig 2C and 2D). The analysis for genes associated with the downregulated repressive peaks regions was inconclusive due to the small gene count and lack of statistical power.

Next, we examined whether age-dependent changes in the histone marks correlate with gene expression changes in the *glp-1* mutants. As a control, we first analyzed our published H3K4me3 peaks and RNA-seq data in the young and old *glp-1* mutants [33]. We examined the genes that were differentially expressed at old age (FDR < 0.01) and overlapped with significantly upregulated and downregulated H3K4me3 peaks. We found a strong positive correlation between RNA expression and H3K4me3 changes (Pearson's correlation coefficient = 0.72;

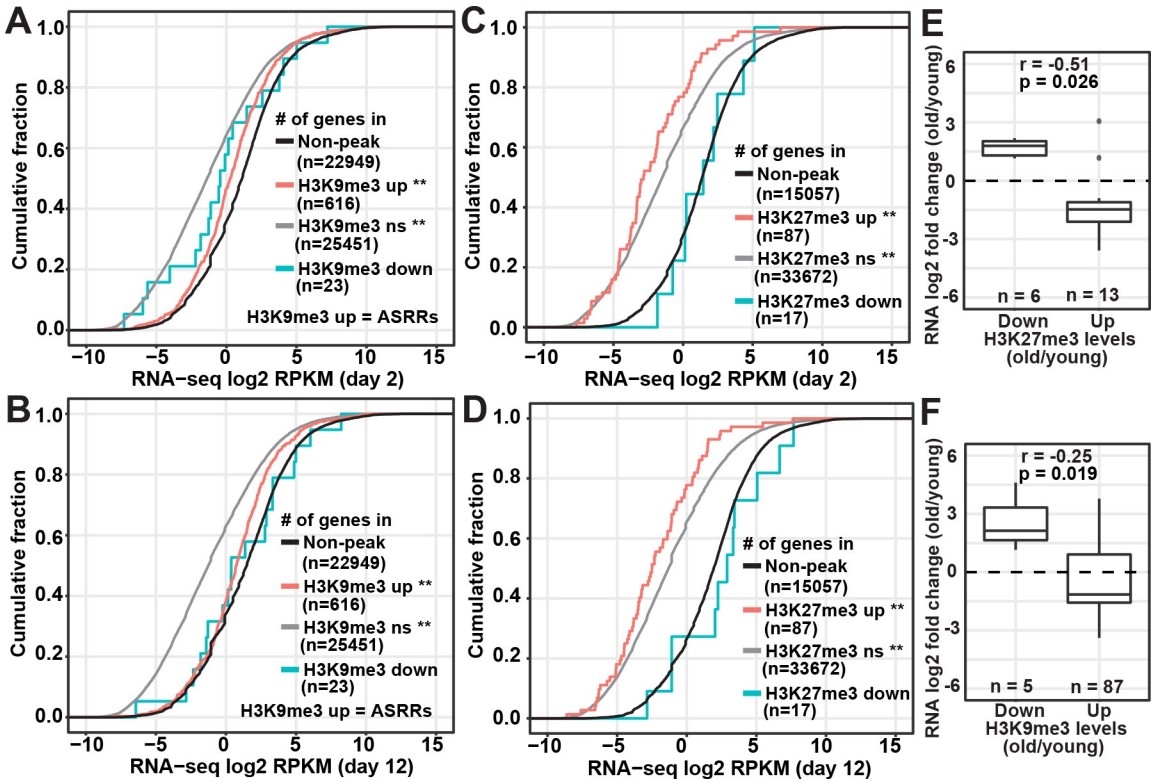

**Fig 2. Anti-correlation between changes in repressive marks and gene expression.** (A-D) Cumulative frequency plots of the normalized RNA read counts (log$_2$ RPKM) of genes associated with H3K9me3 (A and B) or H3K27me3 (C and D) peak regions versus genes in non-peak regions (black curves) in young (day 2) and old (day 12) germlineless *glp-1(e2141)* mutants. Repressive peak regions were divided into three groups based on age-dependent changes: significantly upregulated (up, red curves), significantly downregulated (down, cyan curves), and non-significant (ns, gray curves). Comparing to genes in the non-peak regions (black curves), genes associated with upregulated and non-significant repressive peak regions had significantly lower RNA expression levels (**, *p*-values of Kolmogorov–Smirnov test < 0.0001). The number of genes associated with a given set of peak regions is indicated in the parenthesis. Abbreviation: ASRRs, aging-specific repressive regions. (E and F) Box plots comparing age-dependent RNA expression fold change (log$_2$) of differentially expressed genes associated with significantly upregulated or downregulated H3K27me3 (E) or H3K9me3 (F) peaks. The numbers of differentially expressed genes (n) associated with individual groups of peak regions are indicated in the figures: 87 and 5 genes associated with the 578 upregulated and 17 downregulated H3K9me3 peaks, and 13 and 6 genes associated with the 59 upregulated and 7 downregulated H3K27me3 peaks, respectively. Pearson's correlation coefficient (r) and *p*-value of the correlation between fold changes in RNA expression and histone modifications are indicated on the top of each panel.

S6C Fig), consistent with our previous findings [33]. Next, we examined the differentially expressed genes that overlapped with differential H3K9me3 and H3K27me3 peak regions. We found a strong negative correlation between changes in H3K27me3 and gene expression (Pearson's correlation coefficient = −0.51; Fig 2E). The age-dependent changes in H3K9me3 only weakly anti-correlated with gene expression changes (Pearson's correlation coefficient = −0.25; Fig 2F). It is however important to note that H3K9me3, and H3K27me3, might influence the expression of genes whose coding sequences do not directly overlap with their peak regions, which would be difficult to assess based on the available data.

## New H3K9me3-marked repressive domains were formed in aged somatic tissues

To gain more insights into the genes associated with differential repressive peak regions, we used WormCat to identify their enriched gene ontology (GO) terms [41] and WormBase enrichment analysis tool to identify their tissue specificity and phenotype clusters [42]. We

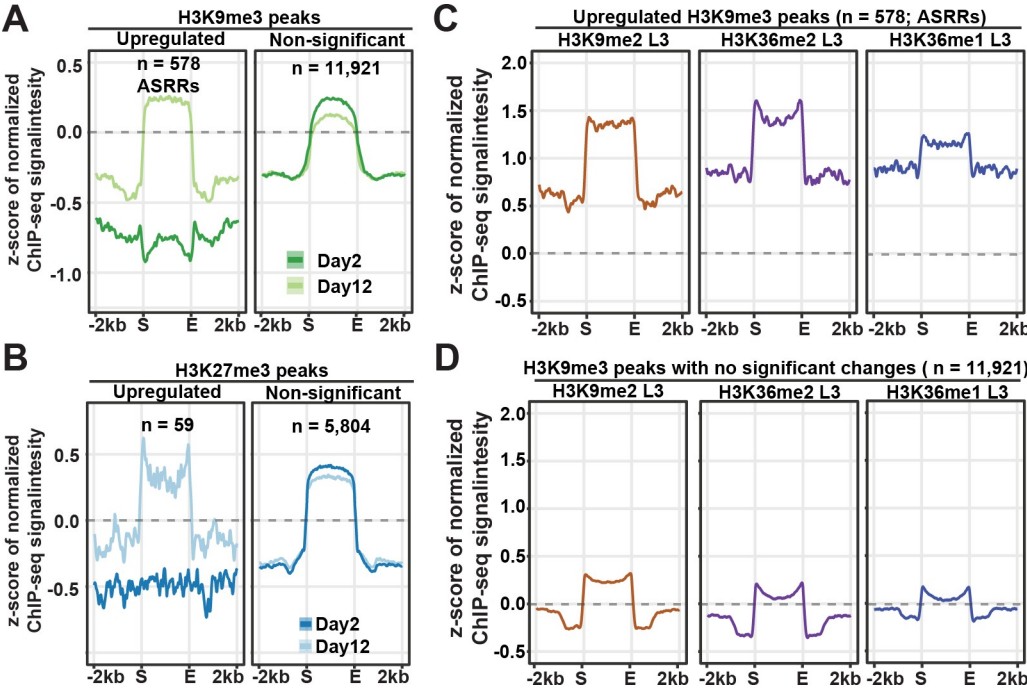

**Fig 3. Age-dependent gain of H3K9me3 in regions marked by H3K9me2 and H3K36me1/2.** (A-B) Metaplots showing the average z-scores of ChIP-seq signal intensity normalized to H3 ($\log_2$) in H3K9me3 (A) or H3K27me3 (B) peak regions (± 2kb flanking regions) with significant gain or non-significant change with age. (C-D) Metaplots showing the z-scores of H3K9me2 and H3K36me1/2 ChIP-seq signal intensity in peak regions with significant gain (C) or no significant change (D) in H3K9me3 levels during aging. The grey dashed lines indicate the z-score of zero.

found no significant enrichment in tissue types and phenotypes for genes associated with downregulated H3K9me3 and H3K27me3 peak regions. Hereafter, we focused on the genes associated with the upregulated H3K9me3 and H3K27me3 peak regions. In the GO enrichment analysis, upregulated H3K9me3 peaks were associated with genes overrepresented for the functional terms related to fundamental cellular processes, particularly "chromatin" (S6D Fig). These genes were enriched for cell cycle phenotypes (S6E Fig) and the tissue germline (S6F Fig). Taken together, the GO data suggested that chromatin genes, many with key roles in the cell division cycle and germline development, preferentially accumulate higher levels of H3K9me3 with age.

While most genes associated with repressive H3K9me3 (n = 12,516) and H3K27me3 (n = 5,870) peak regions were silent, we found that genes associated with significantly upregulated repressive peak regions tends to be actively expressed at both young and old time points in our previously published RNA-seq data (82.0% of the genes in the 578 upregulated H3K9me3 peaks and 58.6% in the 59 upregulated H3K27me3 peaks) [33]. One possibility was that the regions that showed statistically significant upregulation of H3K9me3 and H3K27me3 in aged worms had very low levels of the repressive marks in young worms, so the genes in those regions were actively expressed at the young stage. Indeed, the average plots showed that the upregulated repressive peak regions were deprived of H3K9me3 and H3K27me3 marks at young age (Fig 3A and 3B). At old age, these regions accumulated H3K9me3 and H3K27me3 at levels comparable to the average of the non-differential peak regions (Fig 3A and 3B). This result indicated that certain genomic regions harboring actively expressed genes formed repressive domains by accumulating H3K9me3 and H3K27me3 in aged somatic tissues in *glp-*

*1* mutants. For clarity purposes, the 578 H3K9me3 peak regions newly formed at old age were hereafter referred to as aging-specific repressive regions (ASRRs).

## Age-dependent gain of H3K9me3 in genomic domains marked by H3K9me2 and H3K36me1/2 at young stage

We discussed earlier a blurring of the boundaries between H3K9me3 peak regions and active histone marks (H3K4me3 and H3K36me3) with age (Fig 1C). Next, we examined the levels of active histone marks in upregulated, downregulated, or non-differential H3K9me3 peak regions (S7 Fig). In both upregulated (ASRRs) and non-differential H3K9me3 peak regions, we found the blurred boundaries and spreading of H3K4me3 into H3K9me3 peak regions at old age. Furthermore, unlike the depletion of H3K36me3 in non-differential H3K9me3 peak regions, upregulated H3K9me3 peak regions (ASRRs) exhibited high levels of H3K36me3 at both young and old age. This finding suggested the possibility that the differential repressive peak regions are likely associated with distinct histone modifications and protein binding profiles. This prompted us to examine whether age-dependent changes in H3K9me3 and H3K27me3 were associated with the presence of other histone marks or binding of specific protein factors. We compared the differential repressive peak regions with the peak profiles of publicly available ChIP-ChIP, ChIP-seq, and ATAC-seq data (S6 Table). We included data for 26 histone marks, 268 transcription/chromatin factors, and one ATAC-seq analysis [28,33,35,43–48]. We first used correlation analysis to detect whether two peak profiles were distributed and clustered in similar regions in the genome. We compared the genome-wide distribution of the H3K9me3 or H3K27me3 peaks that changed significantly with age with the peak profile of each of the 295 available datasets by calculating the peak coverage in 50kb sliding windows across the entire genome and performing pair-wise Person's correlation analysis (S8 Fig and S7 Table). Next, we used the permutation test to determine if two sets of peaks have more frequent overlap than expected and computed pair-wise z-scores (S8 Fig and S8 Table). We looked for top-ranked peak profiles in both tests when compared with differential H3K9me3 and H3K27me3 peaks during organismal aging. Among all the comparisons, we found that the 578 peak regions with a significant gain in H3K9me3 (ASRRs) strongly correlated with the peak profiles of H3K9me2 and H3K36me1/2 in wild-type L3 larvae and mixed-stage populations (Pearson's correlation coefficient > 0.45 and permutation z-score > 34). As expected, two H3K9me2-binding factors (HPL-2, LIN-61) and the H3K9me1/2 methyltransferase (MET-2) also had high correlation scores (Pearson's correlation coefficient > 0.39 and permutation z-score > 23). The remaining pair-wise correlations were relatively weak in at least one of the two analyses and would not be further discussed here. We also did not uncover any strong correlation for the differential H3K27me3 peaks. It is important to note that the majority of the publicly available data were from wild-type L3 larvae or mixed-stage embryos. Taken together, the results suggested that significant H3K9me3 gain in ASRRs with aging preferentially occurred at genomic regions decorated with H3K9me2 and H3K36me1/2 in juvenile stages.

To better visualize the association of H3K9me2 and H3K36me1/2 with gain in H3K9me3 during aging, we ranked all H3K9me3 peak regions in descending order according to log2 fold change (old/young) in H3-normalized H3K9me3, and plotted the signal intensity of the indicated histone marks in heatmap format (S9A Fig). In S9A Fig, we divided H3K9me3 peak regions into three groups based on the fold change (old/young) (>1.25 in group A, 0.75~1.25 in group B and <0.75 in group C). We found that the peak regions that gained H3K9me3 with age were preferentially associated with higher levels of H3K9me2 and H3K36me1/2 in L3 stage (Group A in S8A Fig). In contrast, H3K9me2 and H3K36me1/2 enrichment were lower or

depleted in peak regions with stable or decreasing H3K9me3 enrichment with age (Group B and C in S8A Fig). Metaplots showed the strong enrichment of H3K9me2 and H3K36me1/2 marks in upregulated H3K9me3 peak regions (Fig 3C) compared to non-differential peak regions (Fig 3D). These results together pointed to increased H3K9me3 marking during aging preferentially occurred at regions enriched for H3K9me2 and H3K36me1/2 at the larval stage. It is plausible that the gaining of H3K9me3 in the aged *glp-1* mutants was a result of age-dependent methylation of H3K9me2 and conversion into H3K9me3 at these specific regions.

Interestingly, Lee et al. [44] recently reported that elevated levels of H3K9me2 at specific regions are associated with a longer lifespan. Comparing their H3K9me2 profiles from mixed stage worms revealed a strong statistically significant overlap between the regions that showed an age-dependent gain in H3K9me3 in our *glp-1* mutant data with the H3K9me2 enriched regions in the long-lived strains (S9B Fig). This correlation led us to hypothesize that the age-dependent tri-methylation of H3K9 at specific genomic regions we observed in the *glp-1* mutant is accompanied by concomitant loss of H3K9me2 marking, which might contribute to the aging phenotype.

## The expression of CELE45 retrotransposon family increased during aging

H3K9me2/3, hallmarks of heterochromatin, are enriched in repeat-rich DNA domains to repress the transcription of repeat elements such as transposons and satellites. In *C. elegans*, genetic mutations compromising H3K9 methylation or heterochromatic factors caused derepression of repeat elements [48,49]. Although the repeat-rich distal arms of autosomes showed increased H3K9me3 enrichment in aged somatic tissues (S5B Fig), a previous study observed a global loss of H3K9me3 abundance during aging in *glp-1* mutants in Western blotting [9]. Therefore, we examined whether repeat elements could become re-activated and overexpressed in aged somatic tissues. To this end, we analyzed the locus-specific expression of 62,331 repetitive DNA elements based on Dfam2.0 annotation in the ribo-minus total RNA-seq data of day2 and day12 *glp-1(e2141)* adults [33,50]. Repetitive sequences in *C. elegans* are distributed and well-defined due to its holocentric chromosomes. We found 2,867 (4.6%) annotated repeat elements had detectable RNA expression in the *glp-1* mutants at either the young or old time points, indicating the majority of the repeats remained silenced. Furthermore, only 19 of the 2,867 active repeats were silenced at young age, indicating repeat expression at old age was not due to age-dependent derepression. Although the relative expression of repeat elements increased (159%) with age (Fig 4A and S9 Table), the overall abundance of repeat RNAs remained relatively low (0.3~0.5% of the transcriptome) in the *glp-1* worms (Fig 4A). Using edgeR analysis, we identified 173 (0.28%) and 75 (0.12%) repeats with significantly elevated and reduced expression, respectively (S10A Fig and S9 Table). Taken together, these results indicated there was no global age-dependent derepression of silenced repeat elements but a mild increase of the repeat RNAs in the transcriptome (+0.2% of the transcriptome) in aged germlineless *glp-1* mutants.

To exclude the possibility that there was overexpression of uncharacterized repeats missing in the *C. elegans* reference genome assembly (ce11/WBcel235), we examined whether the proportion of unmapped RNA reads increased during aging. After subtracting the RNA reads that mapped to annotated repeats, bacteria, and spike-in DNAs, the fraction of unmapped reads in RNA-seq remained stable (~1.75%) at both time points (S10B Fig), confirming no overexpression of unannotated sequences in aged somatic tissues.

Next, we grouped the differentially expressed repeats by type (DNA transposon, retrotransposon, satellite, or unknown) and found an overrepresentation of RNA retrotransposons among the repeats whose RNA expression increased at old age (120 of 173; Fig 4B). We then

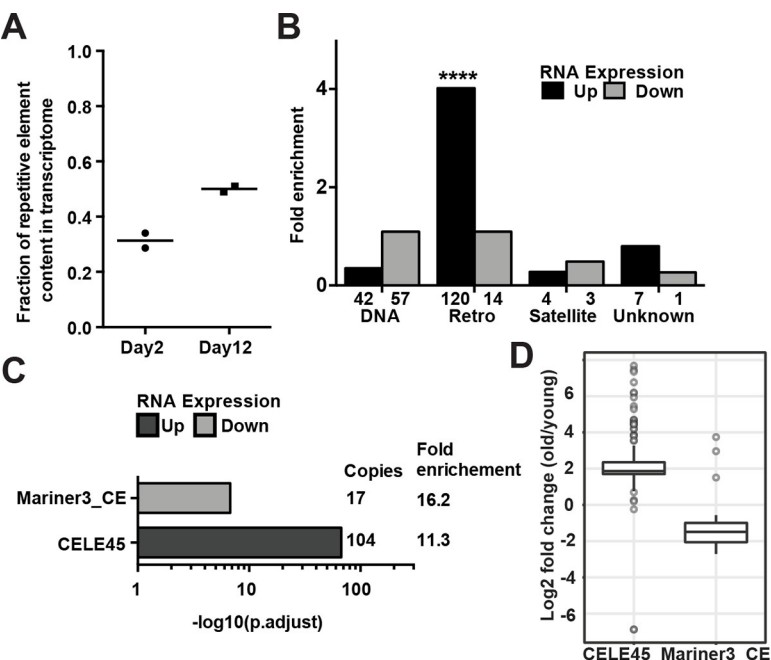

**Fig 4. Overexpression of the putative SINE, CELE45, in aged somatic tissues.** (A) The percentage of mapped RNA reads that originate from repetitive elements significantly increased during aging (two-tailed t-test, *p*-value = 0.023). Horizontal lines represent the mean of two biological replicates of ribo-depleted RNA-seq in young (day 2) and old (day 12) germline-less *glp-1(e2141)* worms. (B) Bar graphs showing the number of repetitive DNA elements in each group (DNA transposons, retrotransposons, satellite elements and unknown class) (x-axis) that showed significantly increased (Up) or decreased (Down) expression with age. Enrichment analysis revealed a significant over-representation of retrotransposon (adjusted *p*-value = $7.14 \times 10^{-68}$) among the repeats with significantly elevated expression. (C) Significant enrichment of CELE45 (104 copies) and Mariner3_CE (17 copies) in differentially expressed repeats in aged somatic tissues. (D) The box and whisker plots show RNA fold changes (log$_2$) of all expressed copies of CELE45 (150 copies) and Mariner3_CE (40 copies) during aging.

grouped the repeats by family and found the DNA transposon family Mariner3_CE and the retrotransposon family CELE45 (Fig 4C) to be overrepresented among the repeats that showed significant age-dependent decrease or increase, respectively, in RNA expression. The Mariner3_CE family represented a tiny fraction of the expressed repeats, and the Mariner3_CE copies appeared to be distributed on all the chromosomes (Figs 4D and S10C). Strikingly, the CELE45 family, which was already well-expressed in young worms, represented ~12.3% of the expressed REs in the old *glp-1* mutants. The CELE45 family is annotated to have 583 copies distributed throughout the genome, with two dense clusters on chromosome X. Our data indicated that most of the actively expressed CELE45 copies (143 of 152) showed a 2-fold or more increase in RNA expression with age, with a somewhat greater induction for the clusters on chr X (Figs 4D and S10D). Since we assigned multi-mapped reads to every matching CELE45 copy, we were concerned about falsely assigning a read from an expressed CELE45 copy to a silent copy that shares high sequence similarity. To address this issue, we performed a similar analysis but only counted uniquely mapped reads, which represented about 31–37% of the CELE45-mapping reads. This analysis revealed that most of the 56 detectably expressed CELE45 copies showed increased expression during aging (23 were statistically significant, with FDR < 0.05; S10E Fig), supporting that a subset of CELE45 copies distributed across the genome showed age-dependent increased expression in the *glp-1* mutants. Since CELE45 is related to the SINE family of retrotransposons, which is known to be transcribed by RNA polymerase III (pol III), we next examined whether the actively expressed CELE45 copies were

bound by RNA pol III in *C. elegans*, using the published ChIP-seq data of the RNA polymerase III subunit RPC-1 in wild-type L3 stage (modENCODE 6300). We found most of the active CELE45 copies (95%) were bound by Pol III, whereas about one-third of the silenced copies (36%) showed Pol III binding (S11A Fig), suggesting that most of the actively expressed CELE45 copies in *glp-1* adults had been actively transcribed starting at larval stages.

To answer whether expression changes of CELE45 and Mariner3_CE copies correlated with age-dependent alterations in repressive marks, we compared the log2 fold changes in RNA and ChIP-seq signals with age. Surprisingly, changes in CELE45 and Mariner3_CE RNA expression did not correlate with changes in H3K9me3 and H3K27me3 marks during aging (S11B and S11E Fig). Instead, actively expressed CELE45 copies with significantly increased expression at old age were marked with high levels of H3K9me3 and H3K27me3 at both young and old age (S11F and S11G Fig). These results indicated that upregulation of CELE45 with aging was not due to derepression by loss of repressive marks.

### CELE45 RNA levels increased in response to stress

We were intrigued by the age-dependent overexpression of the CELE45 in aged somatic tissues in *glp-1* mutants. CELE45 is a putative short interspersed nuclear element (SINE). In mammals, SINEs are rapidly induced in response to a variety of cellular stresses, including heat shock and virus infection [29–31]. To this end, we examined whether CELE45 is induced after heat shock in two previously published RNA-seq data in wild-type *C. elegans* [51,52]. Using the same mapping and differential analyses strategies we applied to our *glp-1* aging data, we found that a large number of CELE45 copies showed significantly elevated expression after heat stress in the two independent published data sets [51,52]. In one data set, we found RNA expression of 59 CELE45 copies rapidly increased after short-term heat shock (30 minutes at 33˚C; S9 Table; Brunquell et al. 2016 [51]). In the other data, we found 49 CELE45 copies showed significantly increased expression after 4 hours at 35˚C (S9 Table; Schreiner et al. 2019 [52]). These results indicated CELE45 was induced in response to heat stress. The age-dependent induction of CELE45 we observed in the *glp-1* mutant might result from age-related cellular stress.

In this study, we used germlineless *glp-1(e2141)* worms raised at the non-permissive temperature (25˚C). We asked whether CELE45 induction was a result of lacking germline and/or being cultured at 25C. To this end, we compared the locus-specific expression of repetitive elements between wild-type worms raised at 20˚C and *glp-1(e2141)* raised at 25˚C using published RNA-seq data [53]. We found that the expression of repetitive elements in young adults at both conditions was surprisingly similar with very few differentially expressed repeats (12 repeat loci in S9 Table), indicating that the lack of germline or a culturing temperature between 20C and 25C did not significantly impact repeat expression. This observation supported the notion that CELE45 upregulation was a specific response to aging.

### Discussion

In this study, we had investigated changes of H3K9me3 and H3K27me3 in aged somatic tissues of *C. elegans* by ChIP-seq. We found that with aging, the overall H3K9me3 levels increased at heterochromatic regions in the distal arms of chromosomes and decreased in euchromatic central regions of autosomes (Fig 1E). H3K27me3 peak regions showed similar patterns of age-dependent changes but at a lower magnitude (Fig 1E). Our data also revealed that 578 peak regions showed statistically significant age-dependent gain of H3K9me3 (S6A Fig), and resided in genomic regions not marked by H3K9me3 during the juvenile and young adult stage (Fig 3A), thus representing aging-specific repressive regions (ASRRs). Newly

formed ASRRs at old age were enriched for H3K9me3 mark but not H3K27me3, whereas the majority of heterochromatic regions formed during *C. elegans* development were dually marked by H3K9me3 and H3K27me3. It is important to note that we used whole worm extracts of germlineness *glp-1* mutants for this study, and therefore our data could not distinguish whether changes in H3K9me3 and H3K27me3 occurred in all or a specific subset of aged somatic tissues. Nevertheless, the reproducible and robust gain of H3K9me3 in ASRRs detected by ChIP-seq at old age suggested that it likely occurs in a substantial fraction of the aged somatic tissues. Future studies investigating tissue-specific H3K9me3 profiles through a developmental and aging time course will help address this question.

Most healthy eukaryotic cells have either round- or oval-shaped nuclei. Abnormal nuclear morphology, which can induce gene expression changes [54,55], occurs during natural and premature aging in worms, flies, and humans [6,13,56]. Many previous studies have linked depletion or defects of lamins, the major structural proteins at the nuclear periphery, to the changes in nuclear morphology [57]. However, a recent study provided evidence that altered repressive histone modifications are sufficient to cause nuclear blebbing without detectable changes in lamins [58]. Therefore, it will be important to investigate whether the region-specific gain and loss of repressive marks observed in this study are linked to nuclear abnormalities during aging in future studies.

Our analysis revealed that ASRRs occur in genomic regions marked by high levels of H3K9me2 and H3K36me2 at juvenile stages. H3K9me2 is a histone mark co-localized with heterochromatin tethered to the nuclear lamina [59], whereas H3K36me2 is an active histone mark associated with actively transcribed genes [60]. H3K9me2-marked heterochromatin has been reported to be transcriptionally permissive in yeast, worms, and flies [61–64]. Here, we found that ASRRs preferentially overlap with gene bodies (92%, 529 of 578 ASRRs), and the majority of the overlapped genes are transcriptionally active in somatic tissues [33]. Importantly, our data indicated that the H3K9me3 gain in ASRRs at old age is associated with transcriptional repression (Fig 2B and 2F). Taken together, the reproducible aging-associated H3K9me3 gain and transcriptional suppression in ASRRs may represent biomarkers of somatic aging, which warrant further investigation.

Although H3K9me2 and H3K9me3 are both regarded as repressive histone marks, recent studies using highly specific antibodies found that they are enriched in different genomic regions and exhibit differential patterns of subnuclear localization [59,65]. H3K9me2 is the evolutionarily conserved mark that tethers heterochromatin to the nuclear lamina at the nuclear periphery. H3K9me2 is enriched in peripheral heterochromatin which moves toward the nuclear interior as its H3K9me2 levels reduced [66]. On the contrary, H3K9me3-marked chromatin is distributed from the nuclear periphery to the interior and not enriched in peripheral heterochromatin that directly interacts with peripheral lamina. Therefore, H3K9me3 gain in H3K9me2-marked ASRRs likely reflects aging-associated changes in subnuclear location and chromatin structure of peripheral heterochromatin. Aging-associated alterations in heterochromatin compartments were also indicated by the spreading of active histone marks from the flanking regions into H3K9me3-marked regions, as seen in Fig 1C. Together, these results suggested dysregulation of heterochromatin organization occurs in aged somatic tissues.

Intriguingly, recent studies implicated that H3K9me2 participates in lifespan regulation and is necessary for the lifespan extension induced by specific genetic mutations or stresses in *C. elegans* [44,67]. Reduced levels of H3K9me2 at a subset of H3K9me2 domains are associated with a shorter lifespan [44]. Based on our results and previous observations, we propose the possibility that loss of H3K9me2 and/or gain of H3K9me3 in ASRRs at old age contribute to the aging process by suppressing gene expression locally and/or triggering global chromatin

remodeling. Future experiments to track the dynamic and abundance of H3K9me2/3 and their functional consequences with age would provide substantial new insights into aging biology.

Across species from flies to mammals, aging is associated with the derepression of transposable elements, which appears coupled with a loss of heterochromatin [11,68–72]. Our analysis in *C. elegans* supported that there was an aging-associated increase in the expression of repetitive DNA elements. Nevertheless, almost all repeats that were silent at young age remained silent at old age, indicating no global derepression of repetitive elements in aged somatic tissues in germlineless *glp-1* mutants (S10A Fig). In *C. elegans*, age-dependent overexpression of the retrotransposon *Cer1* (gypsy/Ty3 family) has been reported in germ cells at 15°C [73]. We found no detectable expression of *Cer1* in our experimental condition (somatic tissues at 25°C), as also noted in the previous study [73]. Instead, we found a small subset of repeats were overexpressed at old age with significant over-representation of the putative SINE, CELE45 (Fig 4C). CELE45 was actively transcribed from early embryogenesis [74] to adulthood (Figs 4 and S11A) in *C. elegans*. The aging-associated over-expression of CELE45 was not due to loss of repressive histone marks (S11G Fig) nor a response to the lack of germline in *glp-1(e2141ts)* mutants grown at the non-permissive temperature (S9 Table). In humans and mice, the expression of SINE RNAs is elevated by cellular stresses and implicated in regulating the expression of stress-responsive genes [29–31,75–77]. Although it is still unclear whether CELE45 participates in stress responses, we found that CELE45 was induced after heat shock in the published RNA-seq data [51–53]. Therefore, CELE45 over-expression at old age is likely a regulated process in response to aging-associated cellular stresses. Going forward, it will be very interesting to determine whether CELE45 is induced by other types of stresses (e.g., bacterial infection), whether it regulates the expression of stress-associated genes like mammalian SINEs do, and whether the overexpression of CELE45 has a functional role in the aging process. Together, this will help us to understand the possible impact of CELE45 overexpression in somatic aging.

## Methods

### *C. elegans* strains and growth

*glp-1(e2141)* animals were maintained and propagated at 16°C. To prepare age-synchronized germlineless *glp-1(e2141)* adults for ChIP-seq experiments, embryos were harvested via bleaching and grown at 25°C on 15-cm nematode growth medium (NGM) plates seeded with 2mL of 30X concentrated overnight culture of *E. coli* strain OP50 containing 50 μg/mL carbenicillin and 15 μg/mL tetracycline. Worms were refed with OP50 at day 4 in adulthood. Adult worms (day 2 and day 12) were washed three times with cold M9 buffer and snap-frozen in liquid nitrogen. Worm pellets were stored at -80°C before chromatin immunoprecipitation (ChIP).

### ChIP-seq library preparation

Frozen worm pellets were ground into a fine powder with a mortar and pestle, followed by cross-linking with 1% formaldehyde in PBS for 10 mins at room temperature. Samples were centrifuged at 4,000 *g* for 5 minutes and washed with M9 three times. Afterward, the cross-linked chromatin was resuspended in FA buffer and sonicated with Bioruptor to generate chromatin fragments with the target DNA length of 200 bp. Chromatin extracts were incubated with anti-H3 (abcam, ab1791), anti-H3K9me3 (abcam, ab8898), or anti-H3K27me3 (07–449, Millipore) antibodies overnight at 4°C. The antibody/chromatin complex was precipitated with Protein A-Agarose beads, washed three times in cold FA buffer, and incubated with proteinase K at 65°C overnight to reverse crosslinking. DNA was purified with QIAquick PCR purification kit and used for Illumina sequencing library preparation. Indexed DNA

libraries were sequenced on the Illumina HiSeq (single-end 50-bp) or NextSeq500 (single-end 75-bp) platforms.

Two independent ChIP-seq experiments were performed. The first experiment (exp1) generated two biological replicates of H3 and H3K27me3 ChIP-seq. The second (exp2) generated two biological replicates of H3 and H3K9me3 ChIP-seq.

## ChIP-seq data analysis

Sequencing reads were preprocessed with a quality filter (fastq_quality_filter -q 20 –p 80), and adapter sequences were trimmed using Trim Galore (v0.6.5). Processed reads were mapped to *C. elegans* reference genome (ce11/WBcel235) using bowtie2 (v 2.3.5.1) with the default settings. Mapped reads were extracted, and true multi-mapped reads were removed with samtools (samtools view -hSb -F 4 -q 2). PCR duplicates were removed using samtools (samtools rmdup). For H3K9me3 and H3K27me3 peak identification, biological replicates of the same experiments were merged for broad peak calling using MACS2 (v2.1.0.2; callpeak -t H3K9me3orH3K27me3.bam -c H3.bam—broad—broad-cutoff 0.7 -g ce -B -m 2 50—nomodel—extsize 200) with a permissive q-value cutoff as described [48]. We took a union set of ChIP-seq peaks from all experiments. Neighboring broad peaks were further merged if the gap is less than 5kb and has an overall positive log2 signal ratio of treatments (H3K9me3 or H3K27me3) over the controls (H3). To identify peak regions with differential enrichment between young and old animals, raw read counts in ChIP-seq peak regions were analyzed using edgeR in R environment (FDR < 0.05) with the comparison: (day12.treatment—day12.H3)—(day2.treatment—day2.H3).

To compare the similarity between biological replicates, the number of mapped reads was counted in 15kb sliding windows. Pearson's correlation coefficient was calculated between the coverage tracks of individual replicates. To visualize ChIP-seq enrichment profiles in young and old animals, mapped reads were first converted into bedGraph tracks of the log2 signal ratio of treatments (H3K9me3 or H3K27me3) over the controls (H3) using deeptools (bamCompare—bamfile1 H3K9me3orH3K27me3.bam—bamfile2 H3.bam—binSize 20—operation log2—scaleFactorsMethod readCount—smoothLength 60—extendReads 200 -of bedgraph). Next, the log2 signal ratio in bedGraph tracks was transformed into z-scores. Z-score bedGraph tracks were converted into bigWig tracks using UCSC bedGraphToBigWig program. computeMatrix utility in deeptools was used to calculate a matrix of z-scores from the bigWig tracks for generating metaplots and heatmaps.

## Correlation analysis of peak profiles

To compare the similarity between peak profiles, we compared their genome-wide distribution and frequency of overlap. We evaluated the genome-wide distribution by calculating the peak area (bp) in 50kb sliding windows across the genome of individual peak profiles. Then, we computed the pair-wise Pearson's correlation coefficients between peak coverage matrices. To determine whether the pair-wise overlap frequency is significantly higher than expected by chance, we used regioneR in the R environment to calculate z-score and *p* value by permutation test [78]. We used the function "overlapPermTest" in regioneR with the following arguments (A = peak1, B = peak2, ntimes = 2000, alternative = "auto", genome = ce11, non. overlapping = TRUE, mc.set.seed = FALSE).

## RNA-seq analysis of repetitive DNA elements

Expression of repetitive elements was analyzed in our previously published ribo-minus RNA-seq of young (day 2) and old (day 12) germlineless *glp-1(e2141)* mutants [33]. Repetitive element reference sequences were created by using Dfam2.0 annotation. Adaptor-trimmed reads

were then mapped to the repetitive element reference sequences with bowtie v1.1.2 (-n 1 -a -best -strata). Read counts were assigned to individual repetitive elements using featureCounts in Subread package in R (Liao 2014). Multi-mapped reads were assigned one read count toward each of the mapped features. Repeat elements that were differentially expressed between young and old animals were analyzed by using DESeq2 in the R environment (Benjamini-Hochberg-adjusted $p$-value < 0.01).

## Supporting information

**S1 Fig. Heatmap showing genome-wide correlation of ChIP-seq replicates.** Biological replicates of individual ChIP-seq experiments were highly correlated. Pair-wise Pearson's correlation coefficient was calculated using genome-wide tag coverage in 15kb sliding windows. There were two independent H3 control and treatment ChIP-seq experiments, H3K27me3 (A and C) and H3K9me3 (B and D).
(TIF)

**S2 Fig. Merging neighboring peak regions with a gap showing positive log2 fold enrichment of repressive marks.** (A) IGV browser screenshot showing z-scores of H3K9me3 signal intensity normalized to H3 in day 2 and day 12 adults (top). Neighboring peaks separated by gaps with a positive mean z-score (yellow wedges) were merged together (bottom). The numbers on the top left corner of each track indicate the y-axis range. (B) The number of H3K9me3 (B) or H3K27me3 (C) peak regions after merging neighboring peaks with a varying maximum threshold of gap size.
(TIF)

**S3 Fig. Repressive H3K9me3 and H3K27me3 were enriched in distal chromosome arms.** (A) H3K9me3 and H3K27me3 were enriched in the distal arm regions of chromosomes. IGV browser view showing the z-score of normalized H3K9me3 or H3K27me3 signal intensity in 20-bp bins across the entire genome. The top four tracks showed the ChIP-seq signal intensity normalized to H3 in young (day 2) and old (day 12) *glp-1(e2141)* adults from data sets in this study. The bottom two tracks showed the ChIP-seq signal intensity normalized to input DNA in N2 L3 larvae from data sets generated by the modENCODE (5037 and 5045). Red and blue bars indicate positive and negative, respectively, z-score. The numbers on the top left corner of each track indicate the y-axis range (z-score). Distal arms of autosomes are highlighted by the orange shaded regions. (B) The genome-wide distribution of active (H3K4me3 and H3K36me3) and repressive (H3K9me3 and H3K27me3) histone modifications were often mutually exclusive. Heatmap showing pairwise Pearson's correlation coefficient between the genome-wide distribution of peak regions of active and repressive histone modifications in adult worms (left). IGV browser screenshot showing the z-scores of H3-normalized signal intensity of active and repressive histone marks (right).
(TIF)

**S4 Fig. Genes associated with repressive peaks were expressed at lower levels.** (A-B) Cumulative frequency plots of the normalized RNA read counts (log$_2$ RPKM) of genes associated with repressive peak regions (red curves) or non-peak regions (black curves). There were net negative changes in the RNA expression levels of genes in H3K9me3 (A) or H3K27me3 (B) peak regions (KS test: **, $p$-value < 0.0001).
(TIF)

**S5 Fig. Repressive marks showed distinct patterns of age-dependent changes in the central and distal arm regions of autosomes.** Metaplots showing the normalized ChIP-seq signal intensity (z-score) of individual histone marks (H3K9me3, H3K27me3, H3K4me3 or

H3K36me3) in young (day 2) and old (day 12) germlineless *glp-1(e2141)* mutants. Peak regions in individual panels are H3K9me3 peaks in the central regions of autosomes (A), H3K9me3 peaks in the distal arms of autosomes (B), H3K27me3 peaks in the central regions of autosomes (C), and H3K27me3 peaks in the distal arms of autosomes (D). There was age-dependent loss of repressive marks in the central regions and gain of H3K9me3 in the distal arms of autosomes. (TIF)

**S6 Fig. Gene ontology (GO) analysis and expression changes of genes in differential H3K9me3 or H3K27me3 peaks.** MA-plot showing $\log_2$ fold changes of H3K9me3 (A) and H3K27me3 (B) ChIP-seq read counts in peak regions with age (old/young) in germlineless *glp-1(e2141)* worms. Peaks with significant changes are highlighted in red (Sig), and the rest are colored in grey (NS, not significant). (C) Boxplots of the RNA $\log_2$ fold change of differentially expressed genes associated with differential H3K4me3 peaks. RNA expression changes strongly correlated with H3K4me3 changes (Pearson's correlation coefficient = 0.72). (D-F) Enrichment analyses of functional terms (D), phenotypes (E), and tissue types (F) were performed on the genes associated with differential H3K9me3 and H3K27me3 peak regions. (TIF)

**S7 Fig. Age-dependent changes of active marks in repressive H3K9me3 peak regions.** (A) Metaplots showing the average z-scores of H3K9me3, H3K4me3 and H3K36me3 ChIP-seq signal intensity normalized to H3 ($\log_2$) across H3K9me3 peak regions (± 2kb flanking regions: significantly upregulated (n = 578), significantly downregulated (n = 17) and non-significant (n = 11,921). H3K4me3 marks showed blurred boundaries and spreading into H3K9me3 peak regions at old age. H3K36me3 were depleted in non-significant H3K9me3 peaks but enriched in upregulated H3K9me3 peaks (ASRRs). (TIF)

**S8 Fig. Flowchart showing analysis steps to identify peak profiles that correlate with differential H3K9me3 and H3K27me3 peak regions.** Differential H3K9me3 and H3K27me3 peaks were first compared with the publicly available datasets for their similarity in genome-wide distribution. Peak coverage in 50kb sliding windows was used to calculate pair-wise Pearson's correlation coefficient. Next, we used the permutation test to compute the expected overlapping frequency between peak profiles. A z-score was given as a measure of the strength of association between peak profiles that deviate from expectation. Based on these analyses, upregulated H3K9me3 peak regions were strongly associated with H3K9me2 and H3K36me1/2 peak regions. (TIF)

**S9 Fig. Heatmap showing H3K9me2 and H3K36me1/2 signals in H3K9me3 peak regions.** (A-B) H3K9me3 peaks were ranked by $\log_2$ fold change with age in descending order from top to bottom and separated into three groups based on $\log_2$ fold changes with age (A, > 1.25; B, 0.75~1.25; C <0.75). (A) Heatmap showing normalized ChIP-seq signal intensity (z-score) of H3K9me3 (young and old *glp-1* mutants), H3K9me2 (N2 L3 larvae), and H3K36me1/2 (N2 L3 larvae) in H3K9me3 peak regions. (B) Normalized H3K9me2 ChIP-seq signal intensity (z-score) in N2 or *wdr-5(ok1417)* worms were generated by Lee et al. (10.7554/eLife.48498) and plotted here in heatmap format. ChIP-seq data were collected after thawing and growing worms for < 6 generations (earlyGen), 8–14 generations (midGen) or 20 generations (lateGen). (TIF)

**S10 Fig. Expression analysis of repeat elements at young and old animals.** (A) MA-plot showing $\log_2$ fold changes of RNA expression of annotated repeats (dfam2.0) with age (old/

young) in *glp-1(e2141)* worms. Repeat elements with significant expression changes are highlighted in red (Sig), and the rest are colored in grey (NS, not significant). (B) The percentage of unmapped reads with unidentified origin did not increase with age. (C) The genomic locations and RNA $\log_2$ fold changes with age of individual Mariner3_CE copies with detectable expression. (D and E) The genomic locations and $\log_2$ fold changes of RNA expression with age of individual CELE45 copies with detectable expression when multi-mapping RNA reads were included (D) or excluded (E). (F) The genomic locations of silent CELE45 copies. (TIF)

**S11 Fig. Changes in RNA expression and repressive marks in CELE45 copies.** (A) Heatmap showing the Pol III ChIP-seq signal intensity (z-score) centered on the transcription start site (TSS) of silenced and expressed CELE45 copies. (B-E) Scatter plots showing the $\log_2$ fold changes of RNA expression and ChIP-seq signal intensity with age. Changes in repressive marks did not account for the overexpression of CELE45 copies and the downregulation of Mariner3_CE copies. (F-G) Metaplots showing the average z-scores of H3K9me3 and H3K27me3 ChIP-seq signal intensity normalized to H3 ($\log_2$) in silenced (F) and expressed (G) CELE45 copies. Expressed CELE45 copies were marked by higher levels of H3K9me3 and H3K27me3 in both young and old worms.
(TIF)

**S1 Table. Summary statistics of ChIP-seq data.**
(XLSX)

**S2 Table. H3K9me3 peak regions.**
(XLSX)

**S3 Table. H3K27me3 peak regions.**
(XLSX)

**S4 Table. Differential H3K9me3 peak regions.**
(XLSX)

**S5 Table. Differential H3K27me3 peak regions.**
(XLSX)

**S6 Table. Summary of publicly available ChIP-ChIP, ChIP-seq, and ATAC-seq data sets.**
(XLSX)

**S7 Table. Pair-wise Pearson's correlation coefficients of genome-wide distribution of peak profiles.**
(XLSX)

**S8 Table. Pair-wise overlap analysis between peak profiles by permutation test.**
(XLSX)

**S9 Table. Differential expression analysis of repetitive DNA elements.**
(XLSX)

## Acknowledgments

Many thanks to Dr. Charles Danko (Cornell University) and Dr. Sylvia Fischer (Harvard Medical School) for insightful discussion and expert advice. Dr. Jeff Glaubitz, Dr. Minghui Wang, and Dr. Qi Sun (Cornell Bioinformatics Facility) for advice on data analysis.

## Author Contributions

**Conceptualization:** Cheng-Lin Li, Mintie Pu, Wenke Wang, Siu Sylvia Lee.

**Data curation:** Cheng-Lin Li, Mintie Pu, Wenke Wang.

**Formal analysis:** Cheng-Lin Li.

**Funding acquisition:** Siu Sylvia Lee.

**Investigation:** Cheng-Lin Li, Mintie Pu, Wenke Wang, Amaresh Chaturbedi, Felicity J. Emerson.

**Methodology:** Mintie Pu, Wenke Wang, Amaresh Chaturbedi, Felicity J. Emerson.

**Project administration:** Siu Sylvia Lee.

**Supervision:** Siu Sylvia Lee.

**Validation:** Cheng-Lin Li, Mintie Pu, Wenke Wang.

**Writing – original draft:** Cheng-Lin Li, Siu Sylvia Lee.

**Writing – review & editing:** Cheng-Lin Li, Mintie Pu, Wenke Wang, Amaresh Chaturbedi, Felicity J. Emerson, Siu Sylvia Lee.

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
