## [Decision Letter · Decision Letter 0]

2 Apr 2021

Dear Dr Lee,

Thank you very much for submitting your Research Article entitled 'Locus-specific H3K9me3 gain in aged somatic tissues in Caenorhabditis elegans' to PLOS Genetics.

The manuscript was fully evaluated at the editorial level and by independent peer reviewers. The reviewers appreciated the attention to an important topic but identified some concerns that we ask you address in a revised manuscript

We therefore ask you to modify the manuscript according to the review recommendations. Your revisions should address the specific points made by each reviewer.

[LINK]

Yours sincerely,

Anne Brunet

Guest Editor

PLOS Genetics

Wendy Bickmore

Section Editor: Epigenetics

PLOS Genetics

Reviewer's Responses to Questions

**Comments to the Authors:**

Reviewer #1: Please see attachment.

Reviewer #2: In this work, Li et al use ChIP-seq to evaluate how H3K9me3 and H3K27me3 marks change with aging. These are repressive chromatin marks, and the authors identify several genes that seem to differentially acquire or lose these marks in aged glp-1 mutant animals. The authors compare their data with previous work measuring age-associated changes in expression, and suggest that, as expected, the increase in repressive marks correlates with decreased transcript abundance in older animals. The authors also find that H3K9me3 marks appear in aged animals in loci that are marked with H3K9me2 marks in juvenile animals. Finally, the authors show that expression of the CELE45 repetitive element, a worm SINE, increases with age. However, the age-associated change in CELE45 is not correlated with changes in repressive chromatin marks. This manuscript presents some interesting data but none of the hypotheses generated in this work are supported by experimental tests. As such, this seems a little premature for publication in PLoS Genetics.

Major questions:

All these ChIP-seq experiments were done using glp-1 to focus on somatic changes. glp-1 mutants are long-lived, in addition to lacking a germline. Could it be that the increased longevity of the glp-1 mutants impacts the age-associated changes in chromatin that the authors observe? Is there any evidence that the age-associated changes in glp-1 mutants are recapitulated in other mutants that lack a germline but are not long-lived (glp-4, fer-1, etc)?

Line 159-164 the authors claim that the distribution of repressive marks in glp-1 is similar to wild-type and that the profiles for H3K9me3 and H3K27me3 between glp-1 adults and wild-type L3. They should show the data supporting these arguments, as they do for active marks in Fig S3B.

The authors claim that MDS analysis suggests that there are "substantial" age-dependent differences in H3K9me3, but "relatively minor" age-dependent variations in H3K27me3 samples (Fig 1A,B). It is not clear to me that this is an accurate description of the data, as MDS shows a simplified projection of distances between data in only two dimensions. How did the authors evaluate the statistical significance of these differences to make this inference? In the MDS, does 2 dimensions capture the variance in the samples, and how did the authors evaluate this?

The data in Fig S3 seem to show that age-associated changes mostly change the magnitude of enrichment of both H3 modifications rather than a wholesale redistribution. This aspect of the data is lost when only the difference between young and old (as in Fig 1E) is shown. This could be misleading to a reader, so I think the authors should include Fig S3A in the main text. The authors should also include labels on the Y-axis of FIg 1E (and all similar plots) to indicate the magnitude of enrichment that is being shown.  It would also be helpful to know which changes are statistically significant when data like these are shown. Finally, the authors should also be more circumspect in claiming "redistribution" in the text of the manuscript.

The authors state that "most of the peak regions that lost repressive H3K9me3 or H3K27me3 marks were not statistically significant at FDR cutoff of 0.05" (line 235-236). But then they claim a correlation of increased gene expression at loci with "downregulated" repressive marks (Fig 2). I don't understand how the authors justify looking for correlations between changes in expression with non-significant changes in repressive marks.

In Fig 2, the authors claim a statistically significant difference between curves based on the KS test. However, some of the differences are quite small. There are also large differences in the number of genes included in the two groups (especially in C and D). The authors need to explain how they ensured that the statistical test was sufficiently powered to detect the changes they report. If they perform these tests on independent data sets are the conclusions robust? Does the reciprocal comparison support these conclusions (ie looking for enrichment of H3K27me3 peaks in genes that are up- or down-regulated by RNAseq)? With the data provided it is nearly impossible for a reader to appreciate the robustness or biological significance of this conclusion.

The authors state that "the majority of genes associated with repressive H3K9me3 and H3K27me3 peak regions were silent" but that "significantly upregulated repressive peak regions were preferentially associated with actively expressed genes" (line 276). This sentence is a bit confusing and should probably be edited for clarity; moreover, I can't figure out where they show this with their data? Doesn't this statement contradict the data shown in Fig 2? Also in this paragraph, the authors claim that the fact the 'upregulated' peaks were lacking repressive marks in young animals supports the idea that these genes were expressed in younger animals. This seems like a circular argument. If you start by looking only at "new" peaks then it seems evident that the repressive marks had to be missing in the young animals, by definition. Does the RNAseq data support the assertion these genes are expressed at day 2 and not day 12?

The authors assert that "most of the actively expressed CELE45 copies in glp-1 adults had been actively transcribed starting at larval stages" and later conclude that the upregulation of this SINE is not associate with loss of repressive marks. It's not clear to me that this has anything to do with the age-associated changes in histone methylation that the authors focused on.

Minor points

- Line 25 and 27 - consider changing "significant" to something else in order to avoid perception you are making a statistical argument.

- Several different sentences in abstract start with "interestingly". The authors should edit the text to avoid overuse of conjunctive adverbs.

- The authors should consider deleting the first paragraph of the Introduction. The second sentence of the introduction is vague and starts with a reference ("these epigenetic marks") that is confusing. Which marks? The first sentence is about deterioration of chromatin structures and epigenetic information, so the reference to epigenetic marks doesn't really make sense. The third sentence is redundant with the first sentence, and the last sentence doesn't really add any information.

- The text needs to be edited for grammar and clarity. There are several sentences that are awkward throughout the text. For example, the second sentence of the abstract (line 20) starts with "Dysregulated epigenome has been linked..." seems to be missing an article. I also noticed an over-abundance of conjunctive adverbs and a few places with random changes in tense.

- It may be more correct to say that the dysregulation of repressive heterochromatin is associated (rather than "linked") with aging (line 69).

- Line 89, sentence beginning with "H3K27me3 is involved in..." needs a citation.

- Line 150, what is the "nevertheless" doing in this sentence?

- What do the numbers on the top left of each trace in Fig S3 mean?

- The authors say that H3K27me3 is enriched in H3KL9me3-depleted central regions of chr II and IV, but data in Fig S3 seems to show that there is also more H3K27me3 signal in the middle of chr V. It's not clear why this was not mentioned by the authors.

- The authors merged neighboring peaks if they were within 5 kb (line 152-153). I don't understand the justification for doing this - are there data showing that peaks are generally this large in C. elegans? What is the average size of a peak in the author's data without this manipulation?

- The authors say that the majority of peaks marked with H3K9me3 and H3K27me3 are not expressed (paragraph starting at line 216). It is not clear to me how they did this analysis.

- The authors claim that the observation that there is not much overlap in where age-associated changes in H3K27me3 and H3K9me3 occur suggests that "different mechanisms" contribute to the age-dependent changes. I'm not sure I follow this logic. I would expect that modifications at K27 and K9 would use distinct writers and erasers, regardless of effects of aging, so I'm not sure what "mechanisms" they are referring to here.

- Line 248 there is a reference written as "(2)" in parenthesis instead of brackets; also I don't think the previous findings referred to here should reference to Sen et al.

- Line 248 it seems there is an extra "Fig" in the parentheses.

- In Fig2A the label for "upregulated" is misspelled.

- The authors use "upregulated" and "downregulated" to refer both to gene expression and accumulation or loss of repressive marks, and this gets confusing at points. The authors need to choose another way to refer to changes in ChIP-seq peaks to avoid ambiguity.

- Line 277: the authors refer to genes that are "silent". Do they mean that in RNAseq these transcripts were not detected? 

- it would be useful to know the experimental conditions for the publically available ChIP-seq and ATAC-seq data sets the authors used to compare to the differential repressive peaks (section starting at bottom of pg 12). A table showing life stage, genotype, reference, and other relevant info would be very helpful here.

- Line 537: is "protease K" supposed to read "proteinase K"?

Reviewer #3: In this manuscript “Locus-specific H3K9me3 gain in aged somatic tissues in Caenorhabditis elegans”, the authors conduct comprehensive ChIP-seq analysis in conjunction with the analysis of publicly available datasets to identify genome-wide patterns of heterochromatin alterations [repressive histone 3 (H3) modifications], during aging. They show that, upon aging, H3K27me3 remains largely unaltered, but H3K9me3 is redistributed across the genome to certain gene rich regions that were marked with other chromatin marks during larval development and growth (H3K9me2 and H3K36me2). In addition, their analysis suggests that the increased expression of specific repeat regions of the genome (e.g. CELE45), known to occur during aging, may not be due to the frank de-repression of these regions by the loss of H3K9me3.

These data and thoughtful analyses, together, add significantly to our understanding of changes in chromatin during aging and suggest that, contrary to the expected loss of heterochromatin that was previously thought to occur during aging, changes may be more nuanced and locus specific. In addition, other mechanisms may account for the increased expression of normally repressed genic elements.

Major Comment:

The only major comment I have is that, as the experimental setup utilizes day 2 and day12 glp-1 (e2141) animals grown at 25C and lacking a germline, it would be important to know if changes that occur during normal aging at 20C recapitulate what is observed in these germline-less mutants. This can be done by assaying by ChIP-PCR H3K9/K27 me3 occupancy at a few randomly selected loci (e.g., those in Fig S6G) which display significant changes in glp-1 animals. These data would be informative as they are more true to what happens during normal aging, even though they may be complicated by the presence of germline cells in these wild-type animals.

Minor comments:

1. Lines 24: typo (repeated word).

**Have all data underlying the figures and results presented in the manuscript been provided?**

Reviewer #1: Yes

Reviewer #2: Yes

Reviewer #3: Yes

PLOS authors have the option to publish the peer review history of their article (what does this mean?). If published, this will include your full peer review and any attached files.

Reviewer #1: No

Reviewer #2: No

Reviewer #3: No

---

## [Decision Letter · Decision Letter 1]

17 Aug 2021

Dear Dr Lee,

We are pleased to inform you that your manuscript entitled "Region-specific H3K9me3 gain in aged somatic tissues in Caenorhabditis elegans" has been editorially accepted for publication in PLOS Genetics. Congratulations!

Yours sincerely,

Anne Brunet

Guest Editor

PLOS Genetics

Wendy Bickmore

Section Editor: Epigenetics

PLOS Genetics

Comments from the reviewers (if applicable):

Reviewer's Responses to Questions

**Comments to the Authors:**

Reviewer #1: All reviewer points have been addressed satisfactorily.

Reviewer #3: The authors have satisfactorily revised the original manuscript.

**Have all data underlying the figures and results presented in the manuscript been provided?**

Reviewer #1: Yes

Reviewer #3: Yes

PLOS authors have the option to publish the peer review history of their article (what does this mean?). If published, this will include your full peer review and any attached files.

Reviewer #1: No

Reviewer #3: No

**Data Deposition**

http://datadryad.org/submit?journalID=pgenetics&manu=PGENETICS-D-21-00229R1

**Press Queries**

---

## [Editor Report · Acceptance letter]

7 Sep 2021

PGENETICS-D-21-00229R1 

Region-specific H3K9me3 gain in aged somatic tissues in <i>Caenorhabditis elegans<i> 

Dear Dr Lee, 

We are pleased to inform you that your manuscript entitled "Region-specific H3K9me3 gain in aged somatic tissues in <i>Caenorhabditis elegans<i>" has been formally accepted for publication in PLOS Genetics! Your manuscript is now with our production department and you will be notified of the publication date in due course.

With kind regards,

Katalin Szabo

PLOS Genetics

On behalf of:
